Exploring the occurrence of thioflavin-T-positive insulin amyloid aggregation intermediates

Ziaunys Mantas mantas.ziaunys@gmc.vu.lt mantas.ziaunys@gmail.com
Sakalauskas Andrius
Mikalauskaite Kamile
Smirnovas Vytautas
Institute of Biotechnology, Life Sciences Center, Vilnius University , Vilnius , Lithuania
Permyakov Eugene
Electronic publication date: 2021 Feb 10
Publication date: 2021
Volume: 9
Electronic Location ID: e10918
Received 2020 Nov 5; Accepted 2021 Jan 18
Copyright: ©2021 Ziaunys et al.
Copyright year: 2021
Copyright holder: Ziaunys et al.
License: This is an open access article distributed under the terms of the Creative Commons Attribution License, which permits unrestricted use, distribution, reproduction and adaptation in any medium and for any purpose provided that it is properly attributed. For attribution, the original author(s), title, publication source (PeerJ) and either DOI or URL of the article must be cited.
License URL: https://creativecommons.org/licenses/by/4.0/

Keywords: Insulin aggregation, Thioflavin-T, Amyloid aggregation, Aggregation intermediates, Double-sigmoidal kinetics

Funding: Vilnius University MSF-JM-3 This research is funded by Vilnius University, grant No. MSF-JM-3. The funders had no role in study design, data collection and analysis, decision to publish, or preparation of the manuscript.

==============================
The aggregation of proteins is considered to be the main cause of several neurodegenerative diseases. Despite much progress in amyloid research, the process of fibrillization is still not fully understood, which is one of the main reasons why there are still very few effective treatments available. When the aggregation of insulin, a model amyloidogenic protein, is tracked using thioflavin-T (ThT), an amyloid specific dye, there is an anomalous occurrence of double-sigmoidal aggregation kinetics. Such an event is likely related to the formation of ThT-positive intermediates, which may affect the outcome of both aggregation kinetic data, as well as final fibril structure. In this work we explore insulin fibrillization under conditions, where both normal and double-sigmoidal kinetics are observed and show that, despite their dye-binding properties and random occurrence, the ThT-positive intermediates do not significantly alter the overall aggregation process.

Introduction

Protein aggregation into amyloid fibrils is linked to multiple neurodegenerative disorders, such as Alzheimer’s, Parkinson’s or prion diseases (Knowles, Vendruscolo & Dobson, 2014; Chiti & Dobson, 2017), affecting millions of people worldwide (Isik, 2010). Such protein assembly into insoluble aggregates is still not fully understood, despite the significant effort put into figuring out both the mechanism of aggregation (Meisl et al., 2016; Castello et al., 2017; Biza et al., 2017; Giorgetti et al., 2018; Linse, 2019), as well as the resulting fibril structural aspects (Makin & Serpell, 2005; Fitzpatrick et al., 2013). As a consequence, there are still very few disease modifying drugs available (Mehta et al., 2017; Cummings et al., 2019; Maurer et al., 2018; Park et al., 2020).

The process of amyloid fibril formation consists of multiple microscopic events. The first one being nucleation, a process during which proteins lose their native structure and form a primary aggregation center (Chatani & Yamamoto, 2018). This structure then passes several growth phases, such as elongation (Gurry & Stultz, 2014), assembly into protofibrils (Dolui et al., 2018) and subsequent maturation into fully formed fibrils (Ma et al., 2013; Sidhu et al., 2017). The resulting aggregates are then capable of acting as catalysts for surface-mediated secondary nucleation (Törnquist et al., 2018), as well as fragmenting into smaller fibrils (Nicoud et al., 2015), thus creating new aggregation centers. Different proteins have also been shown to form specific oligomeric species prior to further aggregation processes (Nettleton et al., 2000; Chiti & Dobson, 2006; Danzer et al., 2007; Selivanova & Galzitskaya, 2012; Sengupta, Nilson & Kayed, 2016). Such a large number of possible steps involved in the aggregation process significantly complicates matters and requires extensive research in order to understand and prevent the progress of amyloid-related diseases.

In order to study aggregation reactions in vitro, insulin is often used as a model amyloidogenic protein (Brange et al., 1997). Despite its main application as a treatment for diabetes, insulin is capable of forming amyloid fibrils under acidic or neutral pH at an elevated temperature (Nielsen et al., 2001). This, coupled with its availability, has made it a widely used protein to study both the mechanisms of amyloid formation (Ahmad et al., 2003; Podestà et al., 2006; Malik & Roy, 2011), as well as possible inhibitory compounds (Wang, Dong & Sun, 2012; Malisauskas et al., 2015; Zheng & Lazo, 2018). Even though a large number of experiments have been conducted with insulin under various conditions, new information regarding its fibrillation continues to arise, such as new possible aggregation mechanisms or structural polymorphisms (Sakalauskas, Ziaunys & Smirnovas, 2019; Ratha et al., 2020). A factor that requires further attention is the seemingly random appearance of double-sigmoidal aggregation kinetics when examining the fibrillization of insulin with a fluorescent probe—thioflavin-T (ThT). This phenomenon was examined by Smirnovas & Winter (2008), Grudzielanek, Smirnovas & Winter (2006) and Foderà et al. (2009), where it was shown that the first increase in the double-sigmoidal curve is likely related to the formation of oligomeric intermediate species capable of binding ThT and this event was consistently reproducible only under certain environmental conditions.

ThT is a benzothiazole dye that binds to the beta-sheet grooves of amyloid fibrils and attains a locked conformation (Robbins et al., 2012). This causes a red shift of its excitation and emission wavelengths, as well as a significant increase in fluorescence quantum yield (Gade Malmos et al., 2017; Xue et al., 2017). The fluorescence intensity, binding affinity and maximum excitation/emission wavelengths are highly dependent on the conformation of fibrils and there are even reports of multiple types of binding modes on the same type of aggregate (Sidhu et al., 2018; Ziaunys, Sneideris & Smirnovas, 2020). Despite being widely used as a probe to track amyloid formation, the dye’s fluorescence is not exclusively tied to such fibrillar aggregates, as it has been shown to increase in fluorescence upon binding or being trapped in non-amyloid structures (Singh et al., 2010; Sulatskaya et al., 2018). This, in turn, does not rule out the possibility of the double-sigmoidal kinetics being the result of structures that are not amyloid in nature.

In our research we observed that when human recombinant insulin is aggregated at pH 2.4, there exist both regular sigmoidal fibrilization kinetic curves, as well as double-sigmoidal ones under the same aggregation conditions. Unlike in the previously reported cases (Grudzielanek, Smirnovas & Winter, 2006; Smirnovas & Winter, 2008; Foderà et al., 2009), this occurrence appears to be random even when the same batch of protein is used. The double-sigmoidal curves also possess a different ThT fluorescence intensity at the end of the reaction, which begs the question whether the formation of these anomalous ThT-positive intermediates could yield differently structured fibrils. Small variations in insulin aggregation conditions, such as pH value or protein concentration (Sneideris et al., 2015; Sakalauskas, Ziaunys & Smirnovas, 2019) can cause the formation of distinct conformation aggregates. Since both regular and double-sigmoidal aggregation types exist under the same conditions, this creates an opportunity to explore any possible differences during the whole fibrillization process. In this work we examine a large sample size of insulin aggregation kinetic curves, isolate the double-sigmoidal kinetic samples from regular ones and determine whether there are secondary structure, morphology, ThT binding and seeding propensity differences between them. In addition, both regular and double-sigmoidal aggregation reactions are tracked by scanning ThT fluorescence excitation-emission matrices in order to determine if there are aggregate structural differences from distinct dye binding, such as specific maximum excitation and emission wavelengths or bound-ThT fluorescence intensity (Groenning et al., 2007; Ziaunys & Smirnovas, 2019b; Ziaunys, Sakalauskas & Smirnovas, 2020).

Materials and Methods

Insulin aggregation

Human recombinant insulin powder (Sigma-Aldrich cat. No. 91077C) was dissolved in a 100 mM sodium phosphate buffer (pH 2.4) containing 100 mM NaCl (reaction buffer). ThT (Sigma-Aldrich cat. No. T3516) was dissolved in H2O to a final concentration of ∼12 mM and mixed for 10 min using vigorous agitation, after which the dye solution was filtered through a 0.22 µm pore syringe filter. An aliquot of the ThT stock solution was diluted 200 times and the exact dye concentration was determined by measuring the solution’s absorbance at 412 nm (ε412 = 23250 M−1cm−1). The ThT stock solution was then diluted to a final concentration of 10 mM. The protein solution was then combined with the reaction buffer and a 10 mM ThTsolution to a final protein and ThT concentration of 100 µM (insulin ε280 = 6,335 M−1cm−1, MW=5808 Da) and distributed into 200 µL test tubes (20 µL final volume). These conditions result in both types of aggregation, with a random appearance of double-sigmoidal kinetic curves.

The aggregation kinetics were tracked as previously described (Milto, Michailova & Smirnovas, 2014). In short, sample ThT fluorescence intensity was monitored using a Qiagen Rotorgene Q real-time analyzer at a constant 60 °C temperature with measurements taken every minute. A total of one thousand samples were measured in batches of 36. After the aggregation reaction, the samples were stored at 4 °C.

For seeded aggregation, fibril samples were sonicated for 10 min using a Bandelin Sonopuls ultrasonic homogenizer with a MS73 tip (40% power, with 30 s sonication/30 s rest intervals). Then insulin, ThT and fibril solutions were combined to a final protein concentration of 100 µM, ThT concentration of 100 µM and 1% or 10−5% fibrils (% of total protein mass in solution). The reaction was monitored as in the non-seeded aggregation experiment.

Kinetic data analysis

A first-order derivative was calculated for each sample’s kinetic data, using a 40-point averaging range. The maximum value of the derivative curve corresponds to the rate of aggregation, while its position—to the time at which the rate is highest (tr). Samples which had one clear peak in the first-order derivative were regarded as normal, while ones which were composed of two peaks (regular, high-rate peak and a small, low-rate peak preceding it)—as double-sigmoidal. Data processing was done using Origin 2018 software.

Fluorescence measurements

Each sample was diluted 5 times to 100 µL with the reaction buffer containing 100 µM of ThT and their fluorescence intensity was measured using a Varian Cary Eclipse Fluorescence Spectrophotometer with 440 nm excitation (slit width—5 nm) and 480 nm emission (slit width—5 nm) wavelengths. For each case, three measurements were taken and averaged.

Atomic force microscopy

The samples were separated into two groups based on their aggregation kinetic profiles and mixed to result in a homogenous solution. 30 µL aliquots were deposited on freshly cleaved mica, incubated for 1 min, washed with 1 mL of MilliQ water and dried under airflow. In the case of intermediate aggregates during the double-sigmoidal aggregation, the real-time analyzer was stopped when the aggregation curve reached the first minor plateau. Then the samples were quickly removed and placed on freshly cleaved mica as mentioned earlier. For each condition, three 10 × 10 µm AFM images were recorded as previously described (Sneideris et al., 2019) using a Dimension Icon (Bruker) atomic force microscope, operating in tapping mode with a silicon cantilever Tap300AI-G (40 N m−1, Budget Sensors). High resolution (1,024 × 1,024 pixels) images were flattened and analyzed using Gwyddion 2.5.5 and SPIP 6.7.8. Each fibril’s height was determined by tracing lines perpendicular to the fibril’s axis. Height statistical analysis was conducted by taking into consideration all three repeats for each condition, i.e., a similar number of aggregates were examined in every image.

Fourier-transform infrared spectroscopy

The two sample groups were centrifuged at 10,000 g for 30 min and resuspended in 1 mL of D2O. The centrifugation and resuspension step was repeated 3 times and the final resuspension volume was 0.25 mL. Before measurements, both samples were sonicated using a Bandelin Sonopuls ultrasonic homogenizer with a MS72 tip (20% power and constant sonication for 30 s). Sonication helps to break fibril clumps, which leads to less scattering effects and better-quality FTIR spectra. During sample preparation and measurement, H-D exchange is insignificant, as in the case of insulin fibrils it is very slow (Dzwolak, Loksztejn & Smirnovas, 2006). The spectra were recorded as previously described (Sneideris et al., 2019). In short, the concentrated fibril samples were scanned in near-vacuum conditions (∼2 mbar) at room temperature using a Vertex 80v (Bruker) IR spectrometer. 256 interferograms were averaged for each spectrum. A D2O spectrum was subtracted and the resulting spectra were normalized to the same area of amide I/I’ band (1,700–1,595 cm−1). Data processing was performed using GRAMS software.

ThT fluorescence excitation-emission matrices

The aggregation solution was prepared as described in the insulin aggregation section to a final volume of 3 mL. The solution was then placed in a 10 mm pathlength cuvette, sealed with a plug cap to prevent evaporation and incubated at 60 °C without agitation. EEMs were scanned every 5 min using a Varian Cary Eclipse fluorescence spectrophotometer using an excitation range from 440 to 465 nm and emission range from 475 to 500 nm (excitation and emission slit widths—5 nm, wavelength step—1 nm, scan rate—600 points/min). Due to the analysis being conducted on a transitioning system, the EEM size was optimized to be as minimal as possible to lower the impact of an intensity drift, which results from different concentrations of aggregates present at the start and finish of each scan cycle. This was done by first acquiring a larger EEM, which encompassed the maximum ThT fluorescence zone, then it was narrowed down as much as possible to reduce scan time.

Each EEM was corrected for the inner filter effect caused by 100 µM of ThT as described previously (Ziaunys & Smirnovas, 2019b). In short, the correction was made by using the absorbance spectra of 100 µM non-bound ThT, as it is extremely difficult to account for absorbance changes throughout the entire reaction resulting from ThT becoming bound to fibrils and because the majority of ThT remains non-bound even when all insulin is aggregated (Ziaunys & Smirnovas, 2019b). The EEM “center of mass” was then calculated for the entire EEM after the inner filter correction. This was done to prevent signal noise, caused by light scattering, from affecting the maximum intensity position.

Results

Aggregation kinetics

A large number (n = 1,000) of low volume insulin samples were aggregated under the exact same conditions and their kinetics were tracked by monitoring changes in ThT fluorescence intensity. Analysis of all the data revealed that a majority of samples experienced normal, sigmoidal spontaneous aggregation kinetics, with one rate maximum seen in the first order derivative (Figs. 1A and 1C). A fraction of samples displayed double-sigmoidal kinetics, with two peaks in the first order derivative curve (Figs. 1B and 1D). The first increase in fluorescence intensity of the double-sigmoidal aggregation kinetics occurred roughly 100 min before the second increase and its rate was, on average, nearly 10-fold lower.

Figure 1 Insulin aggregation curves and their derivatives.

Normal, sigmoidal (A) and anomalous, double-sigmoidal (B) insulin aggregation kinetics and their first order derivatives (C, D respectively).

A total of 55 samples possessed such unusual aggregation kinetics, constituting a probability of such an occurrence being at least 5.5% under the tested conditions. 77 samples had a mixed kinetic profile (very small first peak or a large overlap between both peaks), which could not be accurately attributed to either type of aggregation. The normal and double-sigmoidal samples were then separated for further analysis.

In order to determine if there are any links between the rate of aggregation, the time at which this rate is highest and the resulting fluorescence intensity of fibrils, all three factor dependencies were examined. We can see that all three parameters are mostly independent from one another (Figs. 2A–2C). The time at which the aggregation rate is highest, does not influence the rate at which fibril elongation occurs, neither does it change the final fluorescence intensity of the formed fibrils. When we compare these three factors between the normal and double-sigmoidal samples (Figs. 2D–2I), it appears that the double-sigmoidal fibrillization has a slightly lower time at which the maximum aggregation rate is reached (Figs. 2D and 2G) and it has no effect on the rate itself (Figs. 2E and 2H). There is, however, a considerable difference in the final fluorescence intensity distribution (Figs. 2F and 2I). The distribution maximum is more than 10% lower when the aggregation kinetics are double-sigmoidal, suggesting that there are either off-pathway aggregates or a fraction of fibrils possess a different ThT binding mode. Despite this distinction in average fluorescence intensity, a large portion of all three data sets overlap with one another due to a large spread caused by the stochastic nature of non-seeded insulin aggregation (Foderà et al., 2008).

Figure 2 Distribution of maximum insulin aggregation rate, the time at which it is reached ( tr) and final fibril fluorescence intensity.

Dependence of aggregation rate on tr (A), fluorescence intensity on tr (B) and aggregation rate on fluorescence intensity (C). Distribution of tr (D, G), aggregation rate (E, H) and fluorescence intensity (F, I) for normal and double-sigmoidal samples respectively. Color-coded numbers, displayed above peak-fit curves, indicate peak maximum values.

Fibril structure and seeding properties

The fibril samples were examined using atomic force microscopy (AFM), Fourier-transform infrared spectroscopy (FTIR) and used as seeds to examine their rate of self-replication. In the AFM images acquired during the first increase in signal intensity during double-sigmoidal kinetics, we observe small, round oligomeric aggregate species (with most having a height of 1–2 nm) and short protofibrils (0.1–0.5 µm in length) (Fig. 3A, Fig. S1). When compared to a sample obtained before an increase in dye fluorescence is observed (Fig. S1), there are far more aggregates present in the case of ThT-positive intermediates. The ThT-negative samples also contain a higher number of 0.5–1.5 nm height assemblies and very few elongated structures (Figs. S2A–S2C). When the aggregation reactions are concluded, the fibrils are considerably longer and have a greater height, however, there do not seem to be any major differences between both cases, neither visually nor by their height distribution (Figs. 3B–3D, Fig. S1). Due to formation of large aggregate clumps, the fibrils were sonicated to better examine any possible differences in their height. While the height distribution average values are similar, there is a wider spread in the case of the sonicated double-sigmoidal sample (Fig. S3), likely caused by the existence of several smaller fibrils or amorphous aggregates. The FTIR second derivative spectra (Fig. 3E) and seeding kinetics (Fig. 3F) are also nearly identical for both cases, indicating that the double-sigmoidal aggregation does not have a significant influence on the final fibril secondary structure or self-replication properties.

Figure 3 Normal, intermediate and double-sigmoidal sample AFM images, fibril height distributions, second order FTIR spectra and seeding kinetics.

AFM images of insulin aggregates during the first part of the double-sigmoidal kinetics (A) and at the end of normal (B) and double-sigmoidal (C) aggregation. Height distribution of double-sigmoidal aggregation intermediates and fibrils after normal and double-sigmoidal aggregation (D), where box plots indicate the interquartile range and error bars are one standard deviation. Second order FTIR spectra (E) and seeded aggregation kinetics (F).

ThT-positive intermediates

The normal and double-sigmoidal aggregation reactions were examined by scanning excitation-emission matrices of ThT fluorescence during aggregation. The kinetic curves were plotted as the maximum EEM signal intensity over time. When examining the kinetics of double-sigmoidal aggregation, we see that the lag phase is followed by a slow increase in ThT fluorescence intensity, then a sudden jump in intensity (marked as *), which then quickly returns to a low value and is continued by the second growth phase (Fig. 4A). Such a jump is not visible in the normal aggregation data (Fig. 4A), nor in any of the previous experiments, where samples were only scanned once a minute. This indicates that it may only be visible for a very short time during the EEM scan. When the excitation and emission wavelengths of the EEM “center of mass” are calculated (Figs. 4B and 4C), we see that there are significant changes in both of these parameters during the first, anomalous increase in ThT intensity, as compared to relatively minor variations in the normal aggregation. The excitation wavelength shifts from 457 nm to 453 nm and then rises back to 454 nm, while the emission intensity shifts between 489 nm and 487 nm and gradually reaches 488 nm. Both wavelengths reach a constant value at roughly the same time after the sudden increase in ThT fluorescence intensity, marking the end of the anomalous phase.

Figure 4 Insulin aggregation kinetics and bound ThT fluorescence EEMs.

Insulin aggregation kinetics monitored by scanning ThT fluorescence EEMs (A), where each data point is the maximum value in the recorded EEM. EEM “center of mass” excitation (B) and emission (C) wavelengths over the course of aggregation. Top intensity values present in the ThT EEMs at different aggregation time points (D) during the first double-sigmoidal increase (darker green areas represent higher intensity zones). The red * symbol indicates the point where there is a sudden jump in ThT fluorescence intensity. Data in part (A) was fit using a Boltzmann’s sigmoidal equation with the anomalous aggregation phase data points omitted from the fitting procedure.

If we examine the top fluorescence intensity value distributions in the EEMs before and after the sudden signal jump (Fig. 4D), there are both significant shifts in the top value positions during the anomalous phase, as well as single, high intensity lines. Such lines can be caused by either a large particle floating past the optical path during a scan or by sudden association and dissociation of a ThT-positive aggregate. After the high intensity jump, once the signal returns to normal, these lines are no longer present in any of the EEMs (as seen after 290 min (Fig. 4D)) and they become nearly identical. Such an occurrence has been observed multiple times throughout the study and it ranged from being mild (Fig. S4) to very extreme (Fig. S5).

Discussion

The large sample size of aggregation reactions shows that under these conditions, the occurrence of such anomalous, double-sigmoidal fibrilization kinetics is both relatively rare (5.5%) and seemingly random. This is unlike the previously reported cases, where a certain set of conditions caused all of the kinetic curves to be double-sigmoidal (Grudzielanek, Smirnovas & Winter, 2006; Smirnovas & Winter, 2008; Foderà et al., 2009). The conditions used in this work allowed to examine how such peculiar kinetics affected the overall aggregation reaction, as both types of fibrillizations were observed. When compared to normal aggregation, this anomalous event occurs roughly 100 min before the second increase, however it does not influence the time at which maximum aggregation rate is reached nor the rate of aggregation and the resulting fibrils have an identical morphology, secondary structure and seeding properties. It is possible that it generates a small population of small, amorphous structures, as hinted by the fibril height range, AFM images and the lower ThT fluorescence intensity distribution. The fact that this anomaly does not influence the overall kinetic parameters (apart from lag time) or fibril structure is a positive aspect, considering that these values are often used to determine the effectiveness of anti-amyloid compounds. However, the random formation of different intermediate species does raise concerns. If these structures can appear in any reaction solution and there is minimal connection between their existence and the overall aggregation process, then there is essentially no way of controlling this event.

Judging from the ThT fluorescence EEMs, it appears that during the anomalous aggregation phase, there exists the formation of ThT-positive intermediates or structures that are capable of trapping and conformationally “locking” the dye molecules, unlike during a normal fibrilization process. AFM images acquired during this phase show small and round aggregates, as well as short protofibril species, with an average height that is much lower than observed for fully formed fibrils. Their ability to bind ThT in a different mode (as identified by the higher excitation and emission wavelengths) suggests that they possess a structure that is distinct from both normal intermediate species, as well as the fully formed fibrils. The differences in these parameters could also be an indicator that there is no actual surface-dye interaction, but rather an entrapment of ThT in the oligomeric structure. The lower ThT fluorescence intensity distribution at the end of the reaction and AFM images also hint at a possibility that some of these structures remain in solution and do not become incorporated into the amyloid structure of normal fibrils.

A possible explanation for the ThT-binding ability of these intermediates may stem from the high intensity lines seen in the ThT fluorescence EEMs. When the sample is continuously scanned to generate an EEM, such lines can only result from either a larger particle floating past the optical path or by a quick association and dissociation of a ThT positive aggregate. Considering that during fibrilization the concentration of large aggregates increases, an event which results in enhanced light scattering and signal noise, we would expect to observe an increasing amount of such high intensity lines, however, this is not the case. Once the first phase of the double-sigmoidal kinetics is concluded, they are no longer seen, which means that they are likely not the result of light scattering from larger aggregate particles. This leaves the hypothesis that ThT-positive intermediates quickly associate and dissociate during this time period. If these intermediates are capable of trapping ThT molecules within their structure, they may not even require a similarity to amyloids, as it has been shown that ThT immobilization can cause an increase in fluorescence intensity (Hutter et al., 2011; Ziaunys & Smirnovas, 2019a). This would also explain the significantly different excitation and emission wavelengths, as these parameters depend highly on the dye’s binding mode.

While these ThT-positive intermediates do not have any major effect on the final fibril structure and most kinetic parameters, they could become an issue when testing potential anti-amyloid compounds. If an inhibitor targets the process of primary nucleation and is specific towards a certain type of intermediate structure, the formation of a different type of aggregate, which incorporates ThT but is not amyloid-like, may not be affected at all and result in an increase in ThT fluorescence intensity. This would then lead to a false interpretation on the effectiveness of the tested compound and negatively affect the drug screening process.

Conclusions

The occurrence of double-sigmoidal kinetics during insulin amyloid aggregation does not influence the final fibril structure or morphology, nor does it change the rate of the main reaction. However, it does result in a lower ThT fluorescence intensity and may be related to the formation of a different type of aggregates. The random variability observed during intermediate oligomer formation may also have a negative impact during potential anti-amyloid drug screenings and lead to false interpretations.

Supplemental Information

Figure S1 Atomic force microscopy images of intermediate aggregates, formed during the first part of the double-sigmoidal aggregation (A, B) and fibrils formed during normal (C, D) and double-sigmoidal (E, F) aggregation

Click here for additional data file.

Figure S2 Atomic force microscopy images of intermediate aggregates, formed before an increase in ThT fluoresence intensity during normal aggregation (A–C)

Aggregate height distribution comparison between intermediates formed during normal and double-sigmoidal aggregation (D). Intermediate aggregates were collected and deposited on freshly cleaved mica before an increase in ThT fluorescence intensity was observed.

Click here for additional data file.

Figure S3 Height distributions (A) and AFM images of fibrils after normal (B) and double-sigmoidal (C) aggregation

AFM images were acquired after samples were sonicated as described in the ‘Materials and Methods’ section.

Click here for additional data file.

Figure S4 Insulin aggregation kinetics and bound ThT fluorescence EEMs during a “mild” appearance of double-sigmoidal kinetics

Insulin aggregation kinetics monitored by scanning ThT fluorescence EEMs (A), where each data point is the maximum value in the recorded EEM. EEM “center of mass” excitation (B) and emission (C) wavelengths over the course of aggregation. Top intensity values present in the ThT EEMs at different aggregation time points (D) during the first double-sigmoidal increase (darker green areas represent higher intensity zones). Data in part (A) was fit using a Boltzmann’s sigmoidal equation with the anomalous aggregation phase data points omitted from the fitting procedure.

Click here for additional data file.

Figure S5 Insulin aggregation kinetics and bound ThT fluorescence EEMs during an “extreme” appearance of double-sigmoidal kinetics

Insulin aggregation kinetics monitored by scanning ThT fluorescence EEMs (A), where each data point is the maximum value in the recorded EEM. EEM “center of mass” excitation (B) and emission (C) wavelengths over the course of aggregation. Top intensity values present in the ThT EEMs at different aggregation time points (D) during the first double-sigmoidal increase (darker green areas represent higher intensity zones). Data in part (A) was fit using a Boltzmann’s sigmoidal equation with the anomalous aggregation phase data points omitted from the fitting procedure.

Click here for additional data file.

Supplemental Information 1 Excitation-emission matrix data

Click here for additional data file.

Supplemental Information 2 Seeded aggregation kinetic data

Click here for additional data file.

Supplemental Information 3 FTIR spectrum data

Click here for additional data file.

Supplemental Information 4 Spontaneous aggregation kinetic data

Click here for additional data file.

Supplemental Information 5 Atomic force microscopy images

Click here for additional data file.

Additional Information and Declarations

Competing Interests

Author Contributions

Data Availability

The authors declare there are no competing interests.

Mantas Ziaunys conceived and designed the experiments, performed the experiments, analyzed the data, prepared figures and/or tables, authored or reviewed drafts of the paper, and approved the final draft.

Andrius Sakalauskas and Kamile Mikalauskaite performed the experiments, authored or reviewed drafts of the paper, and approved the final draft.

Vytautas Smirnovas conceived and designed the experiments, analyzed the data, authored or reviewed drafts of the paper, and approved the final draft.

The following information was supplied regarding data availability:

Data, including spontaneous and seeded aggregation kinetic data, Fourier-transform infrared spectra and excitation-emission matrix files, atomic force microscopy images, are available in the Supplemental Files.

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
