# Peer review of "Exploring the occurrence of thioflavin-T-positive insulin amyloid aggregation intermediates"

_PeerJ, doi:10.7717/peerj.10918_

## Round 0.1 · original submission · Major Revisions

I invite you to change your manuscript according to the reviewers' criticism.

Reviewer 1 ·

Basic reporting

No issues

Experimental design

No issues - except that the experimentation could be expanded (see general comments)

Validity of the findings

Also fine

Additional comments

Smirnovas and coworkers undertake to examine a curious but not very diligently reported phenomenon: deviations from sigmoidal behavior in fibrillation. All scientists working with fibrillation kinetics experience these uncanonical time courses, yet they are (speaking from personal and collegial experience) invariably disregarded as inexplicable outliers that defy closer analysis. From that perspective alone the authors are to be commended for trying to address such elusive phenomena.
Major point: Unfortunately I am not really left much wiser from their labours (and therefore not convinced that I or my colleagues should abandon our dismissive attitudes towards such time profiles). The authors do not show any major effect of the non-sigmoidal curves on the fibrillar outcome and do not connect the intermediate species with any specific features. This is actually a pity since it would be interesting to investigate if the intermediates that apparently accumulate midway between the two inflection points have specific properties distinct from what might be observed from simple sigmoidal kinetics. I think the study is of interest for fibrillation researchers simply for following through on this phenomenon – but perhaps mainly to confirm that these phenomena are not particularly important. Unless the authors can show that there are species accumulating along the way of the non-sigmoidal pathways (and not along the sigmoidal one) with distinct properties (e.g. in terms of vesicle permeabilization or cell toxicity etc) it will merely be a calming note telling us not to concern ourselves too much about these aberrations.
Other points:
1. The last sentence of the abstract is not complete. “how it affects…” – what is “it” here and what is the actual conclusion? So far it is just a description of some observations.
2. Introduction line 30: Technically Tafimidis is available against aggregation of transthyretin in Familial amyloidotic polyneuropathy which is considered a neurodegenerative disease.
3. Line 89: “differences from distinct dye binding”. Presumably the authors mean if the ThT fluorescence behavior indicate differences that are not related to dye binding but this should be clarified and some possible alternatives to binding be indicated.
4. The authors should provide details on how the ThT stock solution was prepared – was it simply dissolved in water?
5. Fig. 4A: The y-axis should not be time – that is the x-axis. It is presumably the maximal ThT emission value.
6. Lines 228-230: “Such lines can be caused by either a large particle floating past the optical path during a scan or by sudden association and dissociation of a ThT-positive aggregate.” This sounds extremely stochastical. Is this based on a single observation or multiple ones? It is unclear (both from Results and Discussion) whether the data in Fig. 4 are based on a single double-sigmoidal run or has been observed on multiple occasions.

Reviewer 2 ·

Basic reporting

This manuscript by Ziaunys and co-workers is written clearly, and the Introduction and Background provide proper context for the study. The literature references are appropriate. The structure of the manuscript conforms to the discipline norm. The figures are well made, although I think that the y-axis in panel A of Figure 4 is probably mislabeled (both the x- and y-axes are labeled “Time (min)”). The raw data are supplied. I have two minor comments about the introduction:
a. In the first sentence of the abstract the authors state “The assembly of proteins into highly structured amyloid fibrils is considered to be the main cause of several neurodegenerative diseases”. This statement is highly debatable. Certainly, the process of aggregation seems to drive pathogenesis in many neurodegenerative diseases, but the specific role of amyloid fibrils is disputed as discussed in the Knowles Nat Rev Mol Cell Biol 2014 review that the authors cite. I would suggest changing the wording of this sentence to “The aggregation of proteins is considered to be the main cause of several neurodegenerative diseases”.
b. In lines 29-30 the authors state that there are no disease modifying drugs available for amyloid diseases. However, several drugs are now available to treat the transthyretin amyloidosis (tafamidis, diflunisal, inotersen and patisiran).

Experimental design

The manuscript describes original primary research. The research question, which is to probe the kinetics of amyloid formation by the model protein insulin, is well defined and it is clear how it relates to previous research in the field. The work is generally technically sound. I particularly like the authors’ approach to study the appearance of double sigmoidal kinetics as a rare stochastic event, rather than trying to eliminate it or emphasize it by manipulating experimental conditions. However, I have the following comments:
a. I think that the authors should state clearly the number of distinct samples they examined by AFM to clarify how representative the images are of the contents of their insulin amyloidogenesis reactions.
b. I think the authors should show a representative AFM image of a sample that manifests single sigmoidal kinetics just before the growth phase begins (perhaps around the 250 minute time point) so that it can be compared to AFM image from the equivalent time point in double sigmoidal reactions that is represented in Figure 3A. Such data is needed to understand whether the species formed during the first sigmoidal phase in double sigmoidal aggregation reactions are different from those that exist in single sigmoidal reactions pre-growth phase.
c. The authors should mention what kind of AFM they used- I could not find this information in the Experimental section.
d. I think the authors may be making too much of the spike in fluorescence they observed in the data from the double sigmoidal reaction in Figure 4A, given that it seems like they only observed it in the one sample represented in this figure. The authors need to repeat this experiment, preferably in triplicate at least, or if they were replicated, they should be clear about how many replicates are represented in the figure.
e. The supplemental data for the excitation/emission matrices should be labeled so that it is clear which time points they correspond to.

Validity of the findings

I found the authors results to be sound except for the minor points noted above. Their conclusions are supported by their results and their speculation as to the meaning for the broader field of amyloidosis is appropriate and reasonable.

---

## Round 0.2 · accepted · Accept

I am satisfied by the changes made by the authors.